# *Segatella copri* Outer-Membrane Vesicles Are Internalized by Human Macrophages and Promote a Pro-Inflammatory Profile

**DOI:** 10.3390/ijms26083630

**Published:** 2025-04-11

**Authors:** Alison Sepúlveda-Pontigo, Karissa Chávez-Villacreses, Cristóbal Madrid-Muñoz, Sabrina Conejeros-Lillo, Francisco Parra, Felipe Melo-González, Alejandro Regaldiz, Valentina P. I. González, Isabel Méndez-Pérez, Daniela P. Castillo-Godoy, Jorge A. Soto, Juan A. Fuentes, Katina Schinnerling

**Affiliations:** 1Laboratorio de Inmunología Traslacional, Centro de Investigación de Resiliencia a Pandemias, Facultad de Ciencias de la Vida, Universidad Andrés Bello, Av. República 330, Santiago 8370186, Chile; ali.sepulveda.pontigo@gmail.com (A.S.-P.); karissachavez8@gmail.com (K.C.-V.); cmadridbiochem@gmail.com (C.M.-M.); s.conejeroslillo@uandresbello.edu (S.C.-L.); felipe.melo@unab.cl (F.M.-G.); regaldis.alejandro@gmail.com (A.R.); v.gonzalezhenriquez@uandresbello.edu (V.P.I.G.); i.mndezprez@uandresbello.edu (I.M.-P.); daniela.castillo.g@mail.pucv.cl (D.P.C.-G.); jorge.soto.r@unab.cl (J.A.S.); 2Programa de Doctorado en Biociencias Moleculares, Facultad de Ciencias de la Vida, Universidad Andrés Bello, Santiago 8370186, Chile; 3Laboratorio de Genética y Patogénesis Bacteriana, Centro de Investigación de Resiliencia a Pandemias, Facultad de Ciencias de la Vida, Universidad Andrés Bello, República 330, Santiago 8370186, Chile; f.parralathrop@gmail.com; 4Programa de Doctorado en Biotecnología, Facultad de Ciencias de la Vida, Universidad Andrés Bello, Santiago 8370186, Chile

**Keywords:** *Segatella copri*, outer-membrane vesicles, macrophages, endocytosis, M1/M2 polarization

## Abstract

Increased abundance of *Segatella copri* (*S. copri*) within the gut microbiota is associated with systemic inflammatory diseases, including rheumatoid arthritis. Although outer-membrane vesicles (OMVs) of Gram-negative bacteria are important players in microbiota–host communication, the effect of *S. copri*-derived OMVs on immune cells is unknown. Macrophages engulf and eliminate foreign material and are conditioned by environmental signals to promote either homeostasis or inflammation. Thus, we aimed to explore the impact of *S. copri*-OMVs on human macrophages in vitro, employing THP-1 and monocyte-derived macrophage models. The uptake of DiO-labeled *S. copri*-OMVs into macrophages was monitored by confocal microscopy and flow cytometry. Furthermore, the effect of *S. copri* and *S. copri*-OMVs on the phenotype and cytokine secretion of naïve (M0), pro-inflammatory (M1), and anti-inflammatory (M2) macrophages was analyzed by flow cytometry and ELISA, respectively. We show that *S. copri*-OMVs enter human macrophages through macropinocytosis and clathrin-dependent mechanisms. *S. copri*-OMVs, but not the parental bacterium, induced a dose-dependent increase in the expression of M1-related surface markers in M0 and M2 macrophages and activated the secretion of large amounts of pro-inflammatory cytokines in M1 macrophages. These results highlight an important role of *S. copri*-OMVs in promoting pro-inflammatory macrophage responses, which might contribute to systemic inflammatory diseases.

## 1. Introduction

The interplay between gut microbiota and host immune cells at mucosa is pivotal to the maintenance of intestinal homeostasis, host defense against pathogens, and prevention of inflammation. This host–microbiota interaction can also have profound implications for the development, progression, prevention, and treatment of a wide array of systemic inflammatory and autoimmune diseases that are becoming increasingly prevalent in modern society [1]. The diverse microbial community in the gut shapes host immunity by providing foreign molecular patterns, microbial antigens, metabolites, and secreted factors, which collectively contribute to the development and regulation of immune responses [2].

Among the various components of gut microbiota, outer-membrane vesicles (OMVs) have emerged as key players in the crosstalk between bacteria and host cells [3]. OMVs are nanosized proteoliposomes released from the outer membranes of both pathogenic and commensal Gram-negative bacteria during normal growth [4]. These vesicles enhance bacterial adaptability and survival by facilitating communication, adherence, biofilm formation, antibiotic resistance, and immune modulation [5,6,7]. OMVs contain a diverse array of components, including lipopolysaccharides (LPS), periplasmic and outer-membrane proteins, peptidoglycan, phospholipids, and nucleic acids [4]. However, the exact content depends on the parent bacteria strain and is further influenced by environmental conditions and host-derived factors [8,9]. By delivering virulence factors and short interfering RNAs (sRNAs), OMVs can exert immunomodulatory effects through the activation of pattern recognition receptors or epigenetic regulation of host genes [10]. Despite the growing interest in OMVs, the precise mechanisms by which they influence host immune responses remain inadequately understood. This is particularly significant given their potential to traverse epithelial barriers and target immune cells, such as macrophages, at distal tissues, thereby acting as mediators of systemic inflammatory pathologies [11,12].

Macrophages, with their extensive presence across tissues and organs, are central to innate immune responses due to their phagocytic and immunomodulatory capabilities. They exhibit remarkable phenotypic and functional plasticity, responding to environmental signals by polarizing towards a pro-inflammatory M1 or anti-inflammatory M2 profile. During early immune responses, macrophages adopt an M1 profile, characterized by the expression of major histocompatibility complex (MHC) class II and costimulatory molecules and the secretion of pro-inflammatory cytokines, which promote T-helper (Th)1 and Th17 cell differentiation and can lead to tissue damage if unregulated [13,14]. Conversely, the resolution of inflammation is marked by a shift towards the M2 phenotype, which facilitates tissue remodeling and the maintenance of homeostasis by secreting anti-inflammatory cytokines and favoring the generation of regulatory T (Treg) cells [13,15,16].

*Segatella copri* (formerly known as *Prevotella copri*), an abundant Gram-negative member of the intestinal microbiota, has gained attention due to its controversial role in health and disease [17]. On the one hand, *S. copri* is associated with a fiber-rich diet, and its capacity to metabolize complex plant polysaccharides contributes to the production of short-chain fatty acids (SCFAs) in the gut and improvement of glucose metabolism [18,19,20]. Decreased abundance of *S. copri* has been related to the Western diet and conditions like Parkinson’s disease. On the other hand, *S. copri* enrichment in the gut microbiota is associated with inflammatory pathologies, such as type 2 diabetes, non-alcoholic fatty liver disease, obesity, and rheumatoid arthritis (RA) [21,22,23,24]. Notably, patients with early-onset or preclinical RA exhibit increased levels of *S. copri* in feces, suggesting a potential role in disease pathogenesis [24,25]. In line with this, a considerable proportion of RA patients are characterized by the systemic antibodies and T-cell responses to *S. copri* antigens, and the capacity of *S. copri* to promote experimental arthritis has been demonstrated in different mouse models [26,27,28].

Despite these findings, the mechanisms by which *S. copri* influences immune responses, particularly through OMVs, remain unexplored. This represents a critical gap in the literature, as understanding long-distance communication between *S. copri* and the host via OMVs could elucidate the pathophysiological processes underlying RA and other systemic inflammatory conditions. Previous in vitro and in vivo studies indicate that *S. copri* can degrade the protective mucus layer of the intestinal epithelium and increase the permeability of the epithelial barrier, but the role of its OMVs in these processes is unknown [22,29].

In our laboratory, we have discovered that *S. copri* secretes OMVs; however, their interaction with immune cells, particularly macrophages, has not been characterized. Given the pivotal role of macrophage polarization in chronic inflammatory diseases such as RA, we aimed to investigate the internalization and effect of *S. copri*-OMVs in human macrophages. Specifically, we seek to determine whether *S. copri*-OMVs, compared to the parent bacterium, induce a distinct immune response and influence macrophage polarization in vitro. We found that *S. copri*-OMVs are endocytosed by a portion of macrophages, mainly through macropinocytosis and clathrin-mediated mechanisms. Interestingly, *S. copri*-OMVs, but not the parent bacterium, promote a pro-inflammatory M1-like phenotype and cytokine secretion profile in human macrophages. Moreover, *S. copri*-OMVs sustained the pro-inflammatory state of M1 macrophages and skewed M2-polarized macrophages towards an M1-like profile in vitro. This observation underscores the potential of *S. copri*-OMVs to modulate immune responses in a manner that could exacerbate inflammatory conditions such as RA.

## 2. Results

### 2.1. Segatella copri Secretes Outer-Membrane Vesicles That Are Internalized by Human Macrophages

To explore whether *S. copri* can modulate immune responses through the release of OMVs, we first aimed to purify and characterize *S. copri*-derived OMVs. Following a previously described protocol, we obtained 100 μg of OMVs from supernatants of 250 mL *S. copri* liquid cultures by ultracentrifugation and ultrafiltration [30]. As suggested by Théry et al., we used transmission electron microscopy (TEM) and nanoparticle tracking analysis (NTA) to characterize *S. copri*-derived OMVs [31]. Our analysis identified spherical structures enclosed by a lipid bilayer (Figure 1a) with a size range between 47 and 253 nm (Figure 1b), which are characteristic of OMVs. Quantification of the particle number by NTA and the protein content by Bicinchoninic acid (BCA) assay revealed that 1 μg of protein in preparations of *S. copri*-OMVs corresponds to 2 × 10^8^ OMV particles. Having confirmed that *S. copri* secretes large amounts of nanosized OMVs, these *S. copri*-OMV preparations were used for experiments on human macrophages at concentrations ranging from 0.1 μg/mL to 50 μg/mL, as indicated below.

Internalization of OMVs has been shown for pathogenic and commensal bacteria and might contribute to their effect on host cells [3]. Thus, we assessed the internalization of *S. copri* and *S. copri*-derived OMVs, labeled with the lipophilic green-fluorescent dye DiO, in THP-1-derived macrophages and human naïve macrophages, generated from peripheral blood monocytes (MD-M0), through confocal microscopy. Green-fluorescent bacteria and OMVs were observed on the surface, as well as inside, of THP-1 macrophages and MD-M0 macrophages, as confirmed by orthogonal projection of confocal sections (Figure 1c,d).

To establish the kinetics of internalization of *S. copri*-OMVs, we incubated naïve macrophages derived from the monocytic THP-1 cell line (THP1-M0) with 50 μg/mL of DiO-labeled *S. copri*-OMVs during various time points (0, 15, 30, 60, 120, and 180 min) and measured the percentage of DiO-positive cells and the intensity of the DiO signal by flow cytometry. Internalization of *S. copri*-OMVs in THP-1 macrophages was detectable as early as 15 min after exposure, and the proportion of cells that internalized *S. copri*-OMVs, as well as the amount of internalized *S. copri*-OMVs, steadily increased over time until the measurement endpoint at 180 min (Figure 1e). The total percentage of DiO-positive cells and DiO intensity were superior to the signal detected exclusively within macrophages (after quenching of the surface-associated fluorescence signal by trypan blue), indicating that a proportion of *S. copri*-OMVs adhere to the macrophage surface without being internalized (Figure 1e). The fact that only 28% of THP1-M0 cells were DiO-positive after 180 min indicates that not all macrophages internalized *S. copri*-OMVs. Since heterogenicity of the macrophage population generated from THP-1 cells is rather unlikely, it is more plausible that *S. copri*-OMVs might enter macrophages via different uptake mechanisms.

### 2.2. Segatella copri Outer-Membrane Vesicles Enter Human Macrophages Through Macropinocytosis and Clathrin-Mediated Endocytosis

While bacteria are generally phagocytosed by macrophages [32], nanosized OMVs can use different endocytosis pathways to enter host cells. These include (i) clathrin-dependent endocytosis, which occurs via formation of clathrin-coated pits and can be triggered by ligand binding to cell surface receptors; (ii) caveolin-dependent endocytosis through invagination of cholesterol- and caveolin-enriched microdomains (lipid rafts) in the plasma membrane; and (iii) formation of large actin-driven protrusions from the cell membrane, allowing the uptake of solid particles or liquid from the extracellular space, known as phagocytosis and macropinocytosis, respectively [33]. The choice of route may depend on the size and composition of the OMVs, as well as the interaction with receptors on the host cell membrane [3]. While macropinocytosis allows internalization of particles between 0.2 and 1 μm in diameter, clathrin- and caveolin-mediated endocytosis pathways require cargo smaller than 120 nm and 80 nm, respectively.

To elucidate which endocytosis pathways are utilized for the uptake of *S. copri*-OMVs by macrophages, we blocked endocytic routes with specific inhibitors. Since studies on other Gram-negative bacteria indicate that OMVs mainly enter macrophages via macropinocytosis and clathrin-mediated endocytosis, we focused on these routes [34]. We used Cytochalasin D, which prevents actin polymerization, for the inhibition of macropinocytosis and phagocytosis [35]. Sucrose was applied to block clathrin-dependent mechanisms by removing clathrin-coated complexes from the plasma membrane [36]. THP-1 macrophages were pre-incubated for 1 h with each inhibitor (1 μg/mL Cytochalasin D or 45 mM sucrose), before exposure to 50 μg/mL DiO-labeled *S. copri*-OMVs for a further 3 h. Both inhibitors were shown to be effective by using positive controls: Cytochalasin D blocked phagocytosis of latex beads, while Sucrose blocked the clathrin-mediated internalization of AlexaFluor488-labeled transferrin (Appendix A). Flow cytometry analysis demonstrated a significantly reduced percentage of DiO-positive cells and diminished intensity of the DiO signal after exposure to DiO-labeled *S. copri*-OMVs when Cytochalasin D or sucrose were applied (Figure 1f), indicating that *S. copri*-OMVs enter macrophages by macropinocytosis and, to a lesser extent, by clathrin-dependent endocytosis.

### 2.3. Human M0 Macrophages Acquire M1-like Profile in Response to Segatella copri Outer-Membrane Vesicles

OMV of different bacterial species can exert distinct and even opposite effects on human innate immune cells, inducing either pro-inflammatory or anti-inflammatory responses [37,38,39]. To determine whether stimulation with *S. copri* and its OMVs has pro-inflammatory or anti-inflammatory effects on human macrophages, THP1-M0 were exposed for 24 h to different concentrations of *S. copri* (MOI 10, MOI 30, or MOI 100) or *S. copri*-OMVs (0.1 μg/mL, 1 μg/mL, or 10 μg/mL), and expression of phenotypic markers associated with M1 and M2 profiles was determined by flow cytometry (Figure 2). Live cells were selected for analysis of surface marker expression (Appendix A). An increase in the expression levels of the costimulatory ligands CD80, CD40, and CD86, as well as the antigen-presenting molecule HLA-DR, is a hallmark of pro-inflammatory M1 macrophages. By contrast, an enhanced expression of the mannose receptor CD206 and the scavenger receptor CD163, accounting for an increased phagocytic capacity, is characteristic of anti-inflammatory M2 macrophages. We used Pam3CSK4, a TLR1/2 agonist, as a positive control for the induction of a pro-inflammatory profile. It is of note that neither *S. copri* nor *S. copri*-OMVs induced cell death in THP1-M0 (Appendix A), indicating that *S. copri* and its OMVs are not cytotoxic at the applied concentrations.

To our surprise, THP1-M0 cells stimulated with *S. copri* did not show any significant alteration (displayed as fold increase in MFI) in the expression of M1-related markers in comparison to the control without stimulus (Figure 2a–d). By contrast, *S. copri*-derived OMVs, at a concentration of 1 μg/mL and 10 μg/mL, enhanced the expression of CD80 and CD40, but not CD86, and 10 μg/mL OMVs increased HLA-DR expression in relation to the unstimulated control, comparable to the increase observed with Pam3CSK4 (Figure 2a–d). Concerning M2-related markers, *S. copri*-OMVs, but not the parent bacterium, also decreased CD206 expression, but not CD163, in THP1-M0 (Figure 2e,f). To confirm the M1-like profile induced by *S. copri*-OMVs, we measured the secretion of the pro-inflammatory cytokine TNF-α and anti-inflammatory IL-10 in culture supernatants of THP1-M0 macrophages by ELISA. THP1-M0 macrophages stimulated with *S. copri* secreted neither TNF-α nor IL-10 (Figure 2g,h). However, when THP1-M0 macrophages were stimulated with 1 or 10 μg/mL of *S. copri*-OMVs, an increase in the secretion of TNF-α and IL-10 was observed (Figure 2g,h). This indicates that THP1-M0 macrophages respond to *S. copri*-OMVs, but not to the parental bacterium, by upregulating the expression of M1 phenotypic markers and M1/M2-related cytokines.

Since macrophages derived from the THP-1 cell line have certain limitations—e.g., low expression levels of CD14, rendering them almost unresponsive to bacterial LPS [40], and the requirement of PMA stimulation for macrophage differentiation—we aimed to confirm our results in human macrophages obtained from peripheral blood monocytes of healthy donors. For this purpose, naïve monocyte-derived macrophages (MD-M0) were differentiated by adding M-CSF to the culture media for 8 days. We used the same experimental setting as previously described to determine alterations in the phenotype and cytokine secretion profile of MD-M0 macrophages. We confirmed that neither *S. copri* nor its OMVs induced cell death in MD-M0 cells (Appendix A). Exposure to *S. copri* did not induce any significant alteration in the expression of M1 markers in MD-M0 macrophages in comparison to unstimulated macrophages (Figure 3a–d), while *S. copri*-derived OMVs, already at a low concentration at 0.1 μg/mL, enhanced the expression of CD80 and CD40 in MD-M0 by 2.5 to 3 times with respect to the unstimulated control and comparable to the increase observed with the TLR1/2 ligand Pam3CSK4 (Figure 3a,c). Another M1-related marker, CD86, was also increased in MD-M0 macrophages in comparison to the unstimulated control (Figure 3c). However, the expression of HLA-DR was not significantly altered by stimulation with *S. copri*-derived OMVs (Figure 3d). On the other hand, while *S. copri* did not alter the expression of M2-related markers CD163 and CD206, *S. copri*-derived OMVs at 10 μg/mL induced a decrease in CD163 (but not CD206) expression in MD-M0 macrophages (Figure 3e,f). In accordance with the results obtained in THP-1-derived macrophages, we did not observe alterations in the secretion of pro- and anti-inflammatory cytokines by *S. copri* in MD-M0 (Figure 3h–j). We also did not observe changes in secreted pro-inflammatory IL-6 and IL-23 when MD-M0 were exposed to *S. copri*-OMVs (Figure 3i,j). By contrast, levels of the proinflammatory cytokine TNF-α and the anti-inflammatory cytokine IL-10 in supernatants of MD-M0 macrophages increased in response to stimulation with *S. copri*-OMVs (Figure 3h,k).

Our results indicate that, although *S. copri* is internalized by human macrophages, it fails to induce any pro- or anti-inflammatory response. By contrast, *S. copri*-derived OMVs are potent triggers of M1 polarization in human M0 macrophages, as confirmed by both THP1-M0 and MD-M0 macrophage models.

### 2.4. The Pro-Inflammatory Profile of M1-Polarized Human Macrophages is Strengthened by Segatella copri-Derived Outer-Membrane Vesicles

Considering that strong stimuli may inhibit further activation or even have regulator effects on immune cells [37], we wanted to know whether *S. copri* and its OMVs strengthen or revert M1 polarization in human macrophages. For this purpose, we obtained M1 macrophages by differentiation of human peripheral blood monocytes in the presence of GM-CSF and IFN-γ (Appendix A). These MD-M1 macrophages were exposed for 24 h to different concentrations of *S. copri* (MOI 10, MOI 30, or MOI 100) or *S. copri*-OMVs (0.1 μg/mL, 1 μg/mL, or 10 μg/mL), and the phenotype and cytokine profile were analyzed by flow cytometry and ELISA, respectively. Exposure of MD-M1 macrophages to *S. copri* did not induce any alterations in the expression of M1- or M2-related markers. However, M1 macrophages stimulated with *S. copri*-OMVs exhibited only an augmented expression of the M1 marker CD80 (Figure 4a), without a significant change in the expression levels of M2 markers CD206 and CD163 (Figure 4e,f). Higher doses of *S. copri*-OMVs (1 μg/mL and 10 μg/mL) reduced the viability of MD-M1 macrophages, probably due to activation, while *S. copri* bacteria and OMVs at lower concentrations had no effect on macrophage survival (Appendix A).

When exposed to *S. copri*, MD-M1 macrophages did not secrete any notable levels of pro-inflammatory and anti-inflammatory cytokines. However, when MD-M1 macrophages were stimulated with *S. copri*-OMVs, a dose-dependent significant increase in the secretion of TNF-α, IL-6, IL-23, and IL-10 was observed (Figure 4h–k), being even greater than the increase induced by Pam3CSK4.

These results point to M1 profile-enhancing, cytokine secretion-activating effects of *S. copri*-OMVs, but not of the parental bacterium.

### 2.5. M2-Polarized Human Macrophages Are Repolarized to a M1-like Profile by Segatella copri Outer-Membrane Vesicles

Based on the previous results indicating the strong pro-inflammatory M1-polarizing and M1-strentghening potential of *S. copri*-OMVs, we wondered if human M2-polarized macrophage can be repolarized to an M1 profile by exposure to *S. copri*-OMVs. To this end, human M2 macrophages, differentiated in the presence of M-CSF and IL-4 (Appendix A), were stimulated for 24 h with different concentrations of *S. copri* (MOI 10, MOI 30, or MOI 100) or *S. copri*-OMVs (0.1 μg/mL, 1 μg/mL, or 10 μg/mL), and the phenotype as well as the cytokine profile of these MD-M2 macrophages were determined by flow cytometry and ELISA, respectively. As already shown for MD-M0 and MD-M1 macrophages, MD-M2 macrophages did not alter their expression of M1- or M2-related markers upon exposure to *S. copri* bacteria (Figure 5a–f). In contrast, M2 macrophages stimulated with *S. copri*-OMVs showed a significant increase in the expression of the M1 markers CD40, CD80, CD86, and HLA-DR (Figure 5a–d) but no change in M2 marker (CD163, CD206) expression (Figure 5e,f) in relation to the control without stimulus. Notably, the effect of *S. copri*-OMVs on MD-M2 macrophages was greater than that observed with Pam3CSK4. Neither *S. copri* nor its OMVs induced cell death in MD-M2 macrophages (Appendix A).

Concerning the cytokine profile of MD-M2 macrophages, stimulation with *S. copri* did not alter the secretion of any of the analyzed cytokines, while *S. copri*-OMVs induced a dose-dependent increase in the secretion levels of IL-6 and IL-10 (but not of TNF-α and IL-23), which was significant at a OMV concentration of 10 μg/mL and even more pronounced than the increase stimulated by Pam3CSK4 (Figure 5h–k).

These results point to a M1-repolarizing effect of *S. copri*-OMVs, but not its parent bacterium, on human M2 macrophages.

## 3. Discussion

We demonstrated that the gut bacterium *S. copri* secretes OMVs, which were internalized by human macrophages through macropinocytosis or clathrin-dependent pathways and induce a shift towards a pro-inflammatory M1 profile. Furthermore, *S. copri*-OMVs are strong activators of cytokine secretion in M1-polarized macrophages, able to skew M2-polarized macrophages towards a pro-inflammatory M1-like phenotype.

Secretion of OMVs with a size range from 20 to 350 nm has been described for several Gram-negative bacteria, including pathogens, such as *Helicobacter pylori*, *Salmonella typhimurium*, enterohemorrhagic *Escherichia coli* (EHEC), *Fusobacterium nucleatum*, and *Pseudomonas aeruginosa*, but also for commensal members of the microbiota, such as *Faecalibacterium prausnitzii*, *Bacteroides fragilis*, *Bacteroides thetaiotaomicron*, and *Akkermansia muciniphila*, among others [41,42].

The nanostructures purified from *S. copri* culture supernatants have been confirmed as OMVs by TEM, based on their spherical morphology, integrity, and bilayer structure, similar to OMVs described for other Gram-negative bacteria from gut microbiota, such as *B. fragilis* and *B. thetaiotaomicron* [38,43]. *S. copri*-OMVs were heterogeneous in size, ranging from about 40 nm to 250 nm, as confirmed by TEM and NTA. Smaller OMVs of approximately 47 nm dominate, indicating that both paracellular and transcellular transport through intestinal epithelium may be facilitated, as described for EHEC [44].

We confirmed internalization of both *S. copri* and *S. copri*-derived OMVs, labeled with the lipophilic dye DiO, by human macrophages through confocal microscopy, visualizing fluorescent bacteria and OMVs inside of macrophages. Intracellular *S. copri*-OMVs appeared as green-fluorescent clusters in a portion of macrophages, similar to what was observed for DiO-labeled OMVs from *B. thetaiotaomicron, Enterobacter cloacae,* and *Salmonella* Typhimurium in murine RAW264.7 macrophages [45].

A similar assay had been described for AlexaFluor488-labeled *P. gingivalis*, which was shown to be phagocytosed by RAW 264.7 macrophages [32]. In the same study, M1-polarized macrophages were demonstrated to efficiently kill internalized *P. gingivalis*, while M2 macrophages allowed intracellular survival of *P. gingivalis* [32]. However, *S. copri* is a strict anaerobic gut bacterium, making survival (within macrophages) under normal cell culture conditions very unlikely.

In our kinetics experiments, we determined a steady increase in *S. copri*-OMV internalization in macrophages with time, starting from 15 min and augmenting until at least 180 min after exposure to OMVs. We excluded the possibility of passive transport since performing the assay at 4 °C did not result in any fluorescent signal associated with the cells [46]. The results indicate that *S. copri*-OMVs were internalized more slowly than has been described for OMVs from the gastric pathogen *H. pylori* [47], which is not surprising considering that *S. copri*, as a member of the microbiota, does not contain known virulence factors that facilitate invasion of host cells. Furthermore, to distinguish between macrophages that had internalized *S. copri*-OMVs and those that displayed *S. copri*-OMVs attached to the cell surface, we applied a quenching technique with trypan blue, eliminating fluorescence signals from the cell membrane. This procedure reduced the portion of macrophages containing DiO signal from 35 to 28%, suggesting that *S. copri*-OMVs also adhere and interact with the host cell membrane. The fact that only a portion of human THP-1 macrophages internalized DiO-labeled *S. copri*-OMVs points to different endocytosis mechanisms with distinct kinetics that may be involved in the uptake of *S. copri*-OMVs. However, we cannot completely rule out that membrane labeling of bacteria and OMVs by the lipophilic dye DiO affects interactions with the host cell membrane [33]. Assays with anti-LPS antibodies or other markers present in OMVs could help to address this limitation. Since there are currently no tools available to identify *S. copri*-specific molecules, proteomic and lipidomic analyses of *S. copri*-OMVs are required to unravel biomarkers for their detection and tracking.

Vesicle size can also influence the route of endocytosis in host cells. While smaller vesicles with up to 80 nm and 120 nm are typically internalized via caveolin- and clathrin-dependent mechanisms, larger OMVs can enter cells via macropinocytosis [48,49]. To elucidate the route of endocytosis, we blocked pathways by preincubation with a specific inhibitor: Cytochalasin D for macropinocytosis and sucrose for clathrin-dependent mechanisms. Despite the small medium size of *S. copri*-OMVs, we identified macropinocytosis or clathrin-dependent endocytosis as main mechanisms of uptake by macrophages. This is in accordance with a report on endocytosis of *Brucella abortus* OMVs in THP-1 cells, indicating reduction of macropinocytosis by 42% using cytochalasin D and inhibition of clathrin-mediated endocytosis by up to 33% using monodansylcadaverine [34]. Since endocytosis of *S. copri*-OMVs was not completely blocked by the utilized inhibitors, we cannot exclude the possibility that other mechanisms, such as receptor-mediated endocytosis, or caveolin-dependent or lipid raft-mediated uptake, might contribute to *S. copri*-OMV internalization [33].

Toll-like receptor 4 (TLR4) on the cell surface has been reported to recognize *E. coli* OMVs [50]. OMVs are known to carry LPS and virulence factors that might enable toxin–receptor interactions, facilitating the delivery of vesicle cargo [51]. For example, the adhesin gingipain from *P. gingivalis* binds to host cell receptors and is internalized through clathrin-mediated endocytosis [52]. In the case of *S. copri*-OMVs, it is plausible that the interaction between OMVs and the host cell membrane involves at least LPS-TLR4 engagement [53]. TLR4 stimulation by LPS has been shown to trigger actin remodeling and augment phagocytosis and reactive oxygen species (ROS) production [54,55].

We demonstrated that M0, M1, and M2 macrophages predominantly generate a pro-inflammatory response to *S. copri*-OMVs, but not to the parent bacterium, challenging current knowledge on the effect of *S. copri* on myeloid immune cells [27,29,53]. Identifying the components of *S. copri*-OMVs will be crucial to understanding why *S. copri*-OMVs generate a stronger pro-inflammatory response compared to the parent bacterium. A similar phenomenon has also been described for other Gram-negative bacteria such as *B. fragilis* and its OMVs. While *B. fragilis* activates only TLR4, its OMVs can activate multiple pattern recognition receptors, including TLR2, TLR4, and TLR7, eliciting a stronger innate response in HEK reporter cells [43]. It has been suggested that this effect might be due to selective packaging of proteins, DNA, RNA, LPS, and peptidoglycan in OMVs [43]. OMVs have also been described to enrich toxins and enzymes that promote bacterial virulence. *P. gingivalis*-derived OMVs are selectively enriched with proteases (gingipains), membrane lipoproteins, and LPS [56], while *F. nucleatum* OMVs accumulate outer-membrane proteins, autotransporters with adhesion and cytotoxic functions, the adhesion/invasion protein FadA, and the porin FomA [57]. Thus, selective packaging in OMVs explains why OMVs may elicit a stronger immune response than the parent bacterium [43].

The pro-inflammatory effect induced by *S. copri*-OMVs was even more pronounced than that observed with the positive control Pam3CSK4, a synthetic TLR1/2 ligand. This finding is significant because *S. copri* has been shown to activate the production of pro-inflammatory cytokines primarily through TLR2 in murine bone marrow-derived dendritic cells [58]. Thus, a greater macrophage response to *S. copri*-OMVs compared to the positive control suggests that factors present in the OMVs make them more immunogenic, generating a stronger pro-inflammatory response in immune cells such as macrophages. A detailed characterization of the composition of *S. copri*-OMVs is still pending to be able to elucidate the components in the OMV membrane that exert immunogenic effects.

In M0 macrophages stimulated with *S. copri*-OMVs, a polarization towards a pro-inflammatory M1 profile with an increase in TNF-α and IL-10 secretion was observed. This could be attributed to the high presence of pathogen-associated molecular patterns (PAMPs) such as LPS in OMVs [53]. It has been described that under stress conditions, bacteria increase in the concentration of LPS and envelop proteins that enhance bacterial vesiculation as a form of protection [59]. Stress conditions that favor vesiculation include oxidative stress, nutrient deficiency, temperature, antibiotics, and the presence of oxygen or host immune responses [5]. Therefore, a limitation of this study is that *S. copri* was cultured with anaerobe gas-generating bags, exposing the bacterium to small amounts of oxygen at the beginning of the culture, which might induce stress and increased vesiculation.

It is important to note that macrophages have a high phenotypic plasticity, allowing both M1 and M2 markers to coexist within the same cell [60]. This phenomenon occurs during infections with pathogens, as macrophage bactericidal properties favor M1 polarization. However, pathogens can also mediate M2 polarization to avoid an excessive pro-inflammatory response. Concurrent activating and modulating signals received by the macrophage can lead to dysregulated polarization mechanisms, resulting in an intermediate M1/M2 phenotype during infection [61]. This might explain why both pro-inflammatory and anti-inflammatory cytokines can be secreted simultaneously by the same macrophage.

Our results concerning the effect of *S. copri*-OMVs on human macrophages are consistent with previous studies about murine and human macrophage responses to *P. gingivalis* and its OMVs. These reports have shown that macrophages stimulated with *P. gingivalis* produced low amounts of TNF-α, IL-6, and IL-10, and stimulation with *P. gingivalis* OMVs dramatically increased the secretion of these cytokines [37,62]. It is believed that *P. gingivalis* OMVs act as a decoy, which specifically activate the pro-inflammatory responses in host cells, supporting immune evasion and survival of the parent bacterium [62].

Human M1 macrophages stimulated with *S. copri*-OMVs exhibited a significant increase in the secretion of pro-inflammatory cytokines (IL-23, TNF-α, and IL-6), with levels markedly higher than those in M0 macrophages. We also detected an increase in IL-10 secretion of M1 macrophages, suggesting a counter-regulatory mechanism aimed at mitigating excessive inflammation. Many pro-inflammatory stimuli, such as Gram-negative bacterial LPS—likely present in *S. copri*-OMVs—are known to trigger both a robust inflammatory response and IL-10 production to counterbalance excessive immune activation. This phenomenon has also been described in M1 macrophages stimulated with OMVs from *P. gingivalis* [37]. While M1 macrophages primarily drive inflammation, their ability to produce IL-10 highlights an intrinsic regulatory mechanism to control immune responses. Sustained activation of M1 macrophages can be detrimental to the host, as prolonged inflammation leads to tissue damage [63]. In RA, M1 macrophages accumulate in the synovial membrane, playing a key role in disease progression by acting as antigen-presenting cells, inducing local Th1/Th17 cell responses, and continuously producing large amounts of pro-inflammatory cytokines. This prolonged inflammatory environment contributes to the chronicity and progression of RA, characterized by inflammation of the synovial membrane, pannus formation, and bone resorption [64].

Conversely, M2 macrophages stimulated with *S. copri*-OMVs were repolarized towards a mixed M1/M2 phenotype coexpressing M1- and M2-related markers together with an increased secretion of both pro- and anti-inflammatory cytokines (IL-6 and IL-10). This may be due to the interaction of LPS, or another unknown factor enriched in *S. copri*-OMVs with extracellular or intracellular pattern recognition receptors, which might upregulate transcription factors like IRF5, NF-κB, AP-1, and PPARγ in M2 macrophages, allowing reprogramming to an M1/M2 profile [53,65]. Consequently, it is crucial to identify compounds and possible virulence-associated factors present in *S. copri*-OMVs, which might be responsible for the induction of the observed pro-inflammatory macrophage responses.

We suggest that OMVs may be the missing link between increased *S. copri* abundance in the intestine of RA patients and joint inflammation [24]. Given the increased intestinal permeability observed in RA patients [66], it is conceivable that *S. copri*-OMVs easily cross the intestinal epithelial barrier and interact with macrophages to promote inflammatory responses in the intestine. It has been reported that OMVs of other gut bacteria translocate from mucosal sites to peripheral tissues and even target specific organs [12,67,68,69]. Orally administered *F. nucleatum*-derived OMVs were shown to target joints in a mouse model of collagen-induced arthritis, provoking macrophage-mediated pro-inflammatory responses, which lead to bone and cartilage destruction [12]. Importantly, *S. copri* DNA has been found in the synovial fluid of RA patients, which indicates translocation of *S. copri* components, probably including OMVs, from the gut to the joints [26]. In vivo tracking of *S. copri*-OMVs would reveal which specific organs, including joints, are targeted by these vesicles.

The proinflammatory cytokines IL-23, TNF-α, and IL-6, induced by *S. copri*-OMVs in monocyte-derived macrophages, are related to the generation of IL-17-secreting Th17 cells and a shift in the Th17/Treg balance—a hallmark of autoimmune diseases such as RA [70]. IL-23 binds to IL-23R on T cells, leading to STAT3 phosphorylation and activation of the retinoic acid-related orphan receptor gamma tau (RORγt), which promotes Th17 cell development [71]. TNF-α, on the other hand, inhibits Treg activity by binding to TNFRII, downregulating FoxP3 expression, and reducing its suppressive function [72]. IL-6 induces B-cell maturation, contributes to autoantibody production, Th17-cell proliferation and differentiation, and secretion of monocyte-attracting chemokines by endothelial cells [73]. An increased Th17-cell frequency and IL-17 secretion have been observed in the synovium of RA patients [74]. IL-17, along with TNF-α and IL-6, activates osteoclastogenesis, enabling osteoclasts to adhere to bone surfaces and resorb bone matrix [71,75]. We showed that *S. copri*-OMVs also induces IL-10, an anti-inflammatory cytokine that is increased in RA patients, acting as a counterregulatory mechanism to reduce pro-inflammatory macrophage infiltration, osteoclast activation, and Th17-cell differentiation in the synovium [76]. However, IL-10 immunosuppressive activity is insufficient to inhibit chronic joint inflammation in RA [77]. Given that the inflamed synovial tissue of RA patients is infiltrated by monocyte-derived macrophages [78], we speculate that translocation of *S. copri*-OMVs might activate these cells to produce pro-inflammatory cytokines that drive Th17-cell responses and thereby contribute to disease progression. However, future research, using cocultures of macrophage with T cells, or in vivo experiments are required to prove this hypothesis.

Our findings provide the first evidence that *S. copri* releases OMVs that trigger potent pro-inflammatory responses in human macrophages, which surpass by far the effect of *S. copri* parent bacterium. Since OMVs can act as a vehicle for long-distance delivery of bioactive factors to host cells, the ability of *S. copri*-OMVs to modulate macrophage polarization is highly relevant, not only for the maintenance of gut homeostasis and microbiota–host interaction, but also for the development and progression of systemic inflammatory diseases.

As future directions, the identification of OMV components responsible for the observed pro-inflammatory effect on macrophages is imperative and will also help to reveal possible markers for detection of *S. copri*-OMVs within cells and tissues. Although the stimulation of pattern recognition receptors such as TLR4 by *S. copri*-LPS has been suggested to mediate pro-inflammatory effects, the exact mechanisms by which *S. copri*-OMVs interact with macrophage remain to be elucidated, requiring considering also intracellular NLRs and the action of sRNA [79,80]. Another interesting point to address in the future is whether the internalization of *S. copri*-OMVs, or a specific endocytic route, is required for the observed modulation of macrophage polarization. Finally, in vivo experiments and advanced imaging techniques will be required to track *S. copri*-OMVs and their interaction with macrophages in different tissues.

## 4. Materials and Methods

### 4.1. Human Blood Samples

Seven healthy volunteers (3 female, 4 male) of 23–25 years (median: 24.4 years), not suffering from any significant disorder that requires regular or frequent medication, were recruited, informed about the study, and upon acceptance, signed an informed consent form, approved by the Bioethics Committee of the Faculty of Life Sciences at Andrés Bello University (Approval certificate 001/2023).

### 4.2. Cultivation of Segatella copri

The *Segatella copri* (DSM 18205)-type strain was obtained from the German Collection of Microorganisms and Cell Cultures (DSMZ, *Deutsche Sammlung von Mikroorganismen und Zellkulturen*) and cultured in Brain Heart Infusion (BHI-S) broth, supplemented with 1 g/L cysteine (Merck, Darmstadt, Germany) and 5 g/L yeast extract (BD), at 37 °C under anaerobic conditions, achieved by BD GasPak Anaerobe container system sachets (BD Biosciences, Franklin Lakes, NJ, USA), until the early stationary phase (OD_600nm_ 0.9). The bacterial concentration was displayed in colony-forming units (CFU)/mL, determined by serial dilutions of the overnight culture, obtaining approximately 10^11^ CFU/mL at OD 0.9 and 20 h of culture, as previously reported [58]. For stimulation assays, a bacteria-to-human cell ratio (multiplicity of infection, MOI) of 10, 30, and 100 was used, as indicated.

### 4.3. Purification and Characterization of Outer-Membrane Vesicles

OMVs were isolated as previously described [30]. In brief, *S. copri* was cultured for 20 h in 250 mL of BHI-S until the early stationary phase (OD 0.9). Liquid cultures were transferred to 50 mL centrifuge tubes and centrifuged at 5400× *g* for 30 min at 4 °C. Supernatants were filtered through 0.45 μm filters and then ultrafiltered using 100 kDa Ultracel discs. Concentrated supernatants were ultracentrifuged for 3 h at 150,000× *g* and 4 °C. The sediment containing OMVs was resuspended in Dulbecco’s Phosphate-Buffered Saline (DPBS, Gibco, Waltham, MA, USA) and stored at −80 °C.

For visualization and morphological analysis, OMVs were incubated for 1 min on formvar-coated grids (Electron Microscopy Sciences, Hatfield, PA, USA), the excess was removed with filter paper, and staining was performed with 3% uranyl acetate for 1 min. OMVs were visualized by transmission electron microscopy (TEM) using the Talos F200X G2 (ThermoFisher, Waltham, MA, USA). The OMV number and size distribution were measured through Nanoparticle Tracking Analysis (NTA) using Nanosight NS300 (Malvern Panalytical, Great Malvern, UK). The protein concentration was determined using a BCA assay according to the manufacturer’s instructions (ThermoFisher). We obtained 100 μg of OMVs per 250 mL *S. copri* liquid cultures.

### 4.4. Fluorescent Labeling of Segatella copri and OMVs

For labeling of *S. copri*, bacterial cells from fresh liquid cultures were washed with PBS and adjusted to a concentration of 10^9^ CFU/mL. Staining with 2% *w*/*v* DiO was performed for 30 min in an oxygen-depleted chamber, achieved by the candle jar method. Labeled bacteria were washed with sterile PBS and stored at 4 °C, protected from light or used immediately for internalization assays.

OMVs were stained by adjusting the protein concentration to 500 μg/mL in PBS and the addition of 1% *v*/*v* DiO for 20 min at 37 °C. OMVs were washed three times in sterile PBS and once with serum-free RPMI 1640 medium without phenol red, using previously sterilized 100 kDa Amicon columns (Millipore, Burlington, MA, USA). Centrifuge steps were performed at 4000× *g* for 15 min and 4 °C. Protease inhibitor phenylmethylsulfonyl fluoride (PMSF, 1mM) was added to the resulting suspension, and OMVs were immediately stored at 4 °C, protected from light until use.

### 4.5. Differentiation of THP-1 Macrophages

The human monocytic cell line THP-1 was cultured in RPMI 1640 medium (ThermoFisher), supplemented with 10% fetal bovine serum (FBS; Gibco), 1% L-glutamine, 1% penicillin/streptomycin (Gibco), and 0.05 mM 2-mercaptoethanol, under standard culture conditions at 37 °C with 5% CO_2_. For monocyte differentiation into macrophages, 500,000 cells were seeded in 400 μL medium in 24-well plates and exposed to 100 nM phorbol 12-myristate 13-acetate (PMA) (Stem Cell Technologies, Vancouver, BC, Canada). After 24 h, the medium was replaced with fresh RPMI, and cells were rested for an additional 24 h to minimize potential cytotoxic effects of PMA. Naïve THP-1 macrophages displayed a typical enlarged and adherent morphology (Appendix A) and high viability (Appendix A).

### 4.6. Monocyte Isolation from Peripheral Blood and Differentiation of Monocyte-Derived Macrophages

Peripheral blood mononuclear cells were obtained from fresh heparinized blood samples by density gradient centrifugation using Lymphoprep separation medium (Stemcell Technologies, Vancouver, BC, Canada). Monocytes were isolated using CD14 MicroBeads (Miltenyi Biotec, Bergisch Gladbach, Germany). CD14-positive cells were resuspended in RPMI 1640 medium, with 10% FBS, 1% Penicillin-Streptomycin, and 1% GlutaMAX, and approximately 500,000 cells/500 µL were seeded in 24-well plates.

For monocyte differentiation into macrophages, recombinant human cytokines were added: M-CSF (100 U/mL) for M0 and M2 macrophages, and GM-CSF (640 U/mL) for M1 macrophages. Cytokine-containing medium was replaced every 2–3 days, for a total of 8 days, to obtain naïve M0 macrophages. Subsequently, M1 macrophage differentiation was induced by adding fresh medium with GM-CSF plus IFN-γ (100 U/mL). For M2 macrophage differentiation, M-CSF plus IL-4 (200 U/mL) was added. Monocyte-derived (MD) M0 macrophages were maintained with M-CSF alone for an additional 2 days. Polarized human macrophages adopted a characteristic morphology, as confirmed by light microscopy (Appendix A).

### 4.7. Confocal Microscopy Analysis of the Internalization of S. copri and Its OMVs in Macrophages

Naïve macrophages derived from human monocytes (MD-M0) were cultured at a density of 500.000 cells per well in plates containing previously inserted 12 mm sterile coverslips. DiO-labeled *S. copri* (MOI 100) or OMVs (50 μg/mL) were added for 3 h at 37 °C (control 4 °C). After stimulation, the medium was removed, and cells were washed three times with PBS, fixed with 4% PFA for 20 min, and washed another 3 times with PBS. Macrophages on coverslips were counterstained with membrane dye WGA (2.5 µg/mL) and Hoechst 33,342 (2 µg/mL) nuclear stain for 20 min. Cells were washed 3 more times with PBS, and coverslips were mounted on Fluoromont-G mounting medium (ThermoFisher) on glass slides. Fluorescence images were obtained either on a Leica Sp8 confocal microscope (Wetzlar, Germany) or a Nikon Timelapse microscope (Tokyo, Japan), at 630× magnification.

### 4.8. Assessment of Internalization Kinetics for S. copri-OMVs in Macrophages

To determine the kinetics of internalization, THP-1-derived M0 macrophages were stimulated with either OMVs (50 µg/mL) or *S. copri* (MOI 100) for 180, 120, 60, 30, 15, and 0 min. Cells were washed with PBS, detached with 10 mM EDTA in PBS for 20 min, and resuspended in PBS 0.5% BSA 1 mM EDTA (flow cytometry buffer) for acquisition with a BD FACSymphony A1 flow cytometer (BD Biosciences, Franklin Lakes, NJ, USA). Trypan blue was added to the samples at a ratio of 1:1 to eliminate extracellular fluorescence (quenching) due to attached OMVs.

### 4.9. Inhibition of Endocytosis Pathways

For inhibition of specific endocytosis pathways, THP1-M0 macrophages were seeded in 24-well plates (Corning, Durham, NY, USA) and pre-incubated for 1 h with the respective inhibitor (diluted in supplemented RPMI 1640 medium): 1 µg/mL cytochalasin D (for macropinocytosis) and 45 mM sucrose (for clathrin-dependent endocytosis) or the vehicle DMSO. Then, cells were washed with PBS and challenged for 3 h with 50 µg/mL OMVs. Macrophages were detached with 10 mM EDTA for 20 min at 37 °C, transferred to Eppendorf tubes, centrifuged at 400× *g* for 5 min, and resuspended in flow cytometry staining buffer for sample acquisition using a BD FACSymphony A1 flow cytometer (BD Biosciences, Franklin Lakes, NJ, USA). Fluorescent red-labeled latex beads and 5 mg/mL Alexa Fluor 488-labeled transferrin were used as controls for phagocytosis and the clathrin-mediated pathway, respectively. THP1-M0 cells were pre-incubated for 1 h with the respective inhibitor and subsequently challenged with latex beads for 3 h or transferrin for 15 min, according to the manufacturer’s instructions (ThermoFisher).

### 4.10. Analysis of Macrophage Phenotype by Flow Cytometry

For the analysis of macrophage phenotype, THP1-derived macrophages and human MD macrophages were stimulated for 24 h with 0.5 μg/mL Pam3CSK4 (positive control); *S. copri* at MOI 10, 30, or 100; or *S. copri*-OMVs at protein concentrations of 0.1 μg/mL, 1 μg/mL, or 10 μg/mL. Fluorochrome-labeled antibodies (all from BioLegend, San Diego, CA, USA) against surface markers of pro-inflammatory M1 macrophages, including HLA-DR (PerCP) and CD86 (AlexaFluor488), CD80 (PE-Dazzle594), and CD40 (AlexaFluor647), as well as anti-inflammatory M2 markers like CD163 (PE/Cy7) and CD206 (PE), were used to determine fluorescence intensity (equivalent to relative expression levels) by flow cytometry. Antibodies were diluted 1:100 and added to cells for 20 min at 4 °C in the dark. Samples were then washed and fixed with paraformaldehyde, and the cell pellet was resuspended in flow cytometry buffer (PBS 0.5% BSA 1mM EDTA). Samples were acquired using a FACSymphony A1 flow cytometer and analyzed with FlowJo v.10 software (BD Biosciences, Franklin Lakes, NJ, USA).

### 4.11. Assessment of Cytokine Secretion by ELISA

The concentration of IL-6, IL-23, TNF-α, and IL-10 in cell culture supernatants was measured using sandwich ELISA kits (all from Biolegend), according to the manufacturer’s instructions. Data were corrected by subtracting the basal amount of cytokine secretion by unstimulated macrophages from each experimental condition.

### 4.12. Statistical Analysis

GraphPad Prism 8.0 software (GraphPad Software, Inc., San Diego, CA, USA) was used for statistical analysis. To determine significant differences between data sets, one-way ANOVA with the Dunnett post hoc test was used (* *p* ≤ 0.05, ** *p* ≤ 0.01, *** *p* ≤ 0.001, **** *p* ≤ 0.0001). Data were expressed as mean ± standard error of the mean (SEM).

## 5. Conclusions

*S. copri* secretes OMVs, which enter human macrophages via macropinocytosis and clathrin-mediated endocytosis. Importantly, *S. copri*-derived OMVs exert potent pro-inflammatory effects on human macrophages in vitro, which contrast with the low stimulatory capacity of *S. copri* parent bacterium. The capacity of *S. copri*-OMVs to increase secretion of Th1/17, polarizing cytokines in M1 macrophages, and skew polarization of M2 macrophages towards an M1-like profile could play a decisive role in the context of systemic inflammatory diseases such as RA.

## Figures and Tables

**Figure 1 ijms-26-03630-f001:**
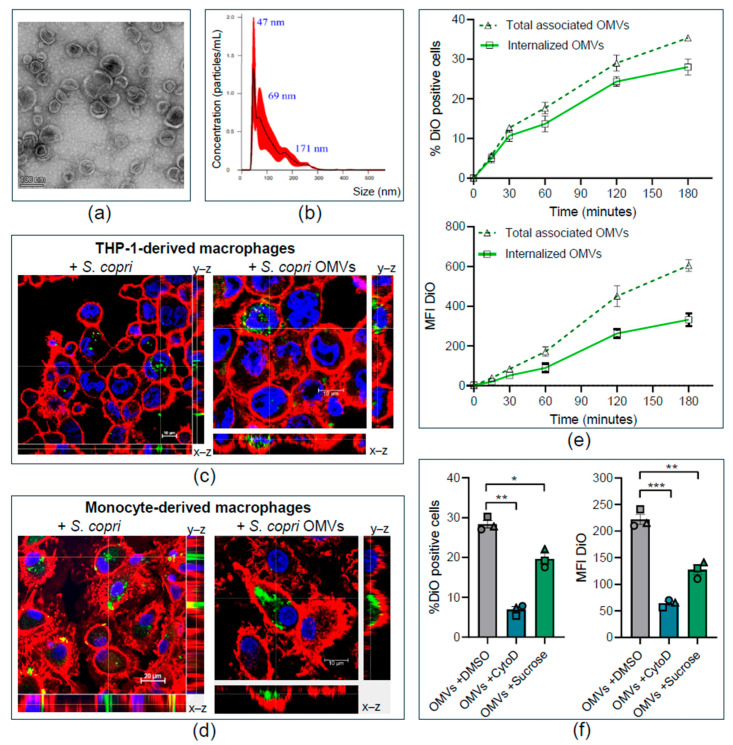
Characterization of *Segatella copri*-derived OMVs and their internalization by human macrophages. (**a**) Transmission electron microscopy (TEM) image of *S. copri*-OMVs, captured at a total magnification of 73,000×. (**b**) Size distribution of *S. copri*-OMVs, measured by nanoparticle tracking analysis (NTA), n = 3. (**c**,**d**) Orthogonal projection of confocal sections 630× of human THP-1-derived macrophages (**c**) and peripheral blood monocyte-derived macrophages with 3× digital zoom (**d**) that were exposed for 3 h to DiO-labeled *S. copri* at MOI 100 (**left**) or DiO-labeled OMVs at 50 µg/mL (**right**). DiO shows green fluorescence; plasma membrane was counterstained with WGA (red) and nuclei with Hoechst 33,342 (blue). (**e**) Kinetics of *S. copri*-OMV uptake by THP-1-derived macrophages over a time of 3 h. Percentage of DiO-positive cells (**upper panel**) and mean fluorescence intensity (MFI) of DiO (**lower panel**), determined by flow cytometry after exposure to 50 µg/mL DiO-labeled *S. copri*-OMVs. Fluorescence was measured before (dotted line) and after the addition of trypan blue (continuous line) to quench the DiO signal of *S. copri*-OMVs adhered to the macrophage membrane. (**f**) Uptake mechanism of *S. copri*-OMVs into macrophages was determined by pre-incubation of THP-1-derived macrophages for 1 h with 1 µg/mL Cytochalasin D, 45 mM sucrose or vehicle (DMSO) before exposure to 50 µg/mL DiO-labeled *S. copri*-OMVs for further 3 h. Data (n = 3) are displayed as mean ± SEM. Each symbol represents an independent experiment. Significant differences, according to one-way ANOVA and the Dunnett multiple comparison test, are indicated (* *p* ≤ 0.05, ** *p* ≤ 0.01, *** *p* ≤ 0.001).

**Figure 2 ijms-26-03630-f002:**
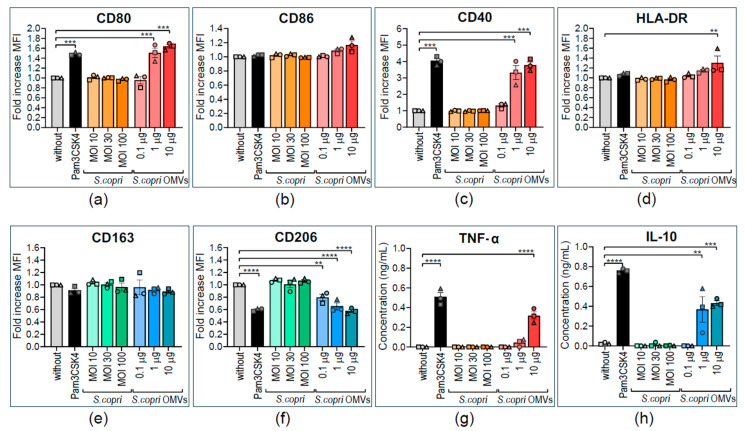
Altered expression of pro- and anti-inflammatory surface markers and cytokines by human THP1-derived macrophages in response to *Segatella copri* and its OMVs. THP-1 macrophages were stimulated for 24 h with Pam3CSK4 (0.5 μg/mL); *S. copri* at multiplicity of infection (MOI) 10, 30, or 100; or *S. copri*-OMVs at 0.1 μg/mL, 1 μg/mL, or 10 μg/mL. (**a**–**c**) Expression of surface markers related to an M1 phenotype, CD80 (**a**), CD86 (**b**), CD40 (**c**), and HLA-DR (**d**), as well as the M2 markers CD163 (**e**) and CD206 (**f**), were determined by flow cytometry. Alterations of mean fluorescent intensity (MFI) with respect to unstimulated control were displayed as fold increase. The pro-inflammatory cytokine TNF-α (**g**) and anti-inflammatory IL-10 (**h**) were quantified in culture supernatants by ELISA. Each symbol represents an independent experiment. All results are expressed as mean ± SEM. Statistical significance was determined by one-way ANOVA and the Dunnett post-test, with ** *p* < 0.01, *** *p* < 0.001, and **** *p* < 0.0001 (n = 3).

**Figure 3 ijms-26-03630-f003:**
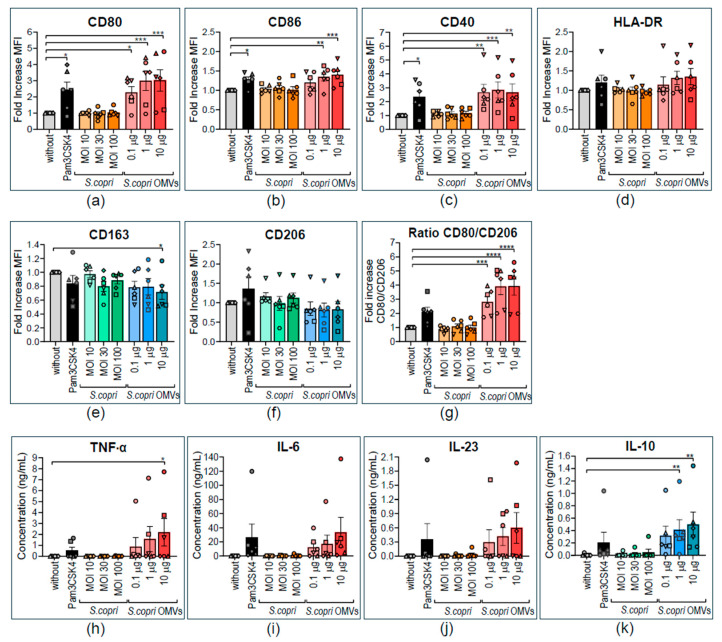
Altered expression of pro- and anti-inflammatory surface markers and cytokines by human monocyte-derived M0 macrophages in response to *Segatella copri* and its OMVs. Peripheral blood monocytes were differentiated into M0 macrophages during 10 days in the presence of 100 U/mL M-CSF and subsequently exposed for a further 24 h to Pam3CSK4 (0.5 μg/mL, positive control); *S. copri* at MOI 10, 30, or 100; or *S. copri*-OMVs at concentrations of 0.1 μg/mL, 1 μg/mL, or 10 μg/mL. Expression of CD80 (**a**), CD86 (**b**), CD40 (**c**), and HLA-DR (**d**) was characteristic of an M1 profile, as well as CD163 (**e**) and CD206 (**f**), M2-related markers, and the CD80-to-CD206 ratio (**g**) was determined by flow cytometry. Alterations of mean fluorescent intensity (MFI) with respect to the control without stimulus were displayed as fold increase. Secretion of the pro-inflammatory cytokines TNF-α (**h**), IL-6 (**i**), and IL-23 (**j**) and anti-inflammatory IL-10 (**k**) was measured in culture supernatants by ELISA. Each symbol represents a different blood donor. Results are displayed as mean ± SEM. Statistical significance was determined by one-way ANOVA and the Dunnett post-test, with * *p* ≤ 0.05, ** *p* ≤ 0.01, *** *p* ≤ 0.001, and **** *p* ≤ 0.0001 (n = 6).

**Figure 4 ijms-26-03630-f004:**
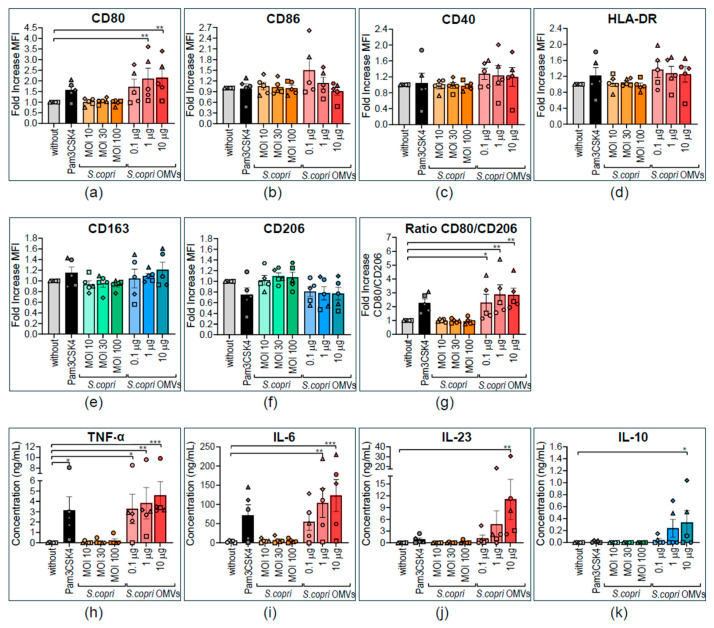
Altered expression of pro- and anti-inflammatory surface markers and cytokines by human M1 macrophages in response to *Segatella copri* and its OMVs. Peripheral blood monocytes were differentiated into macrophages during 8 days in the presence of 1000 U/mL GM-CSF. M1 polarization was induced by adding GM-CSF and IFN-γ (200 U/mL) for the last 2 days of culture. M1 macrophages were exposed to Pam3CSK4 (0.5 μg/mL, positive control); *S. copri* at MOI 10, 30, or 100; or *S. copri*-OMVs at concentrations of 0.1 μg/mL, 1 μg/mL, or 10 μg/mL for a further 24 h, and expression of CD80 (**a**), CD86 (**b**), CD40 (**c**), and HLA-DR (**d**) was characteristic of a M1 profile, as well as CD163 (**e**) and CD206 (**f**), M2-related markers, and the CD80-to-CD206 ratio (**g**) was determined by flow cytometry. Alterations of mean fluorescent intensity (MFI) with respect to the control without stimulus were displayed as fold increase. Secretion of the pro-inflammatory cytokines TNF-α (**h**), IL-6 (**i**), and IL-23 (**j**) and anti-inflammatory IL-10 (**k**) was measured in culture supernatants by ELISA. Each symbol represents a different blood donor. Results are expressed as mean ± SEM. Statistical significance was determined by one-way ANOVA and the Dunnett post-test, with * *p* ≤ 0.05, ** *p* ≤ 0.01, and *** *p* ≤ 0.001 (n = 5).

**Figure 5 ijms-26-03630-f005:**
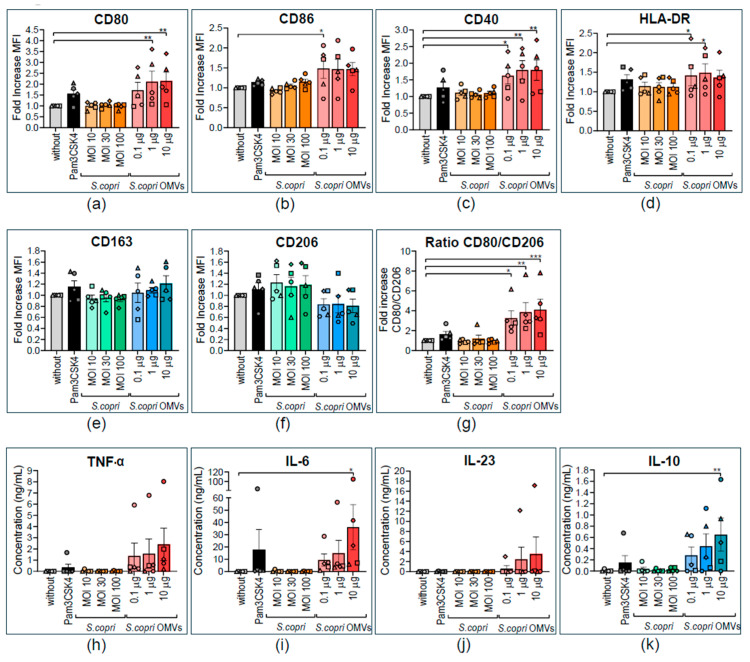
Altered expression of pro- and anti-inflammatory surface markers and cytokines by human M2 macrophages in response to *Segatella copri* and its OMVs. Peripheral blood monocytes were differentiated into macrophages in the presence of 100 U/mL M-CSF, and on day 8, M2 profile was induced by adding M-CSF and IL-4 (20 ng/mL) for a further 48 h. After stimulation with Pam3CSK4 (0.5 μg/mL, positive control); *S. copri* at MOI 10, 30, or 100; or *S. copri*-OMVs at concentrations of 0.1 μg/mL, 1 μg/mL, or 10 μg/mL for 24 h, the surface expression of the M1-related markers CD80 (**a**), CD86 (**b**), CD40 (**c**), and HLA-DR (**d**), and the M2 markers CD163 (**e**) and CD206 (**f**), as well as the CD80/CD206 ratio (**g**), was determined by flow cytometry. Alterations of mean fluorescent intensity (MFI) with respect to the control without stimulus were displayed as fold increase. Secretion of the pro-inflammatory cytokines TNF-α (**h**), IL-6 (**i**), and IL-23 (**j**) and anti-inflammatory IL-10 (**k**) was measured in culture supernatants by ELISA. Each symbol represents a different blood donor. All results are expressed as mean ± SEM. Statistical significance was determined by one-way ANOVA and the Dunnett post-test, with * *p* ≤ 0.05, ** *p* ≤ 0.01, and *** *p* ≤ 0.001 (n = 5).

## Data Availability

The original contributions presented in this study are included in the article/Appendix A. Further inquiries can be directed to the corresponding author.

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
