# Peer review of "Segatella copri Outer-Membrane Vesicles Are Internalized by Human Macrophages and Promote a Pro-Inflammatory Profile"

_ijms, 2025, doi:10.3390/ijms26083630_

Round 1
Reviewer 1 Report
Comments and Suggestions for Authors
This study investigated the effect of Segatella copri OMVs on macrophage polarization using THP-1 cells and human PBMC-derived macrophages. The results indicate that macrophages uptake S. copri OMVs via macropinocytosis and clathrin-dependent endocytosis. S. copri OMVs induce M1 polarization of macrophages.
The author found that 1 μg of S. copri OMVs contains 2 × 10⁸ OMV particles. To obtain 1 μg of S. copri OMVs, what culture volume of S. copri is needed?
Figures 1c and 1d require scale bars and labeling of the side panels for orthogonal projections of confocal sections.
Since Figures 1c and 1d show S. copri internalization, why only present OMV uptake kinetics and not S. copri uptake kinetics? For comparison, S. copri uptake kinetics should also be included.
Figure 1F shows that cytochalasin D and sucrose inhibit S. copri OMV uptake at different levels. It is worth investigating whether the combined use of these inhibitors can achieve complete blockage of S. copri OMV internalization. Furthermore, the effect of cytochalasin D and sucrose on S. copri cell internalization should be examined.
Figure 2. Samples were collected after 24 hours of incubation. Were earlier or later time points examined? Given that uptake kinetics were only measured up to 3 hours, does OMV uptake continue to increase over 24 hours? If uptake plateaus at an earlier time, would the same effects be observed as at 24 hours? Similarly, if uptake continues to increase, would a stronger response be observed?
While Pam3CSK4 effectively induces M1 polarization, the inclusion of a well-established M2 polarization agent as a positive control for M2 polarization would enhance the study's rigor.
It appears S. copri OMVs have different effects on THP-1 cells and PBMC-derived macrophages. To resolve this discrepancy, a comparative analysis of uptake kinetics in both cell types is warranted. Figures 1e and 1f present data exclusively for THP-1 cells, corresponding results for PBMC-derived macrophages should be included. Considering the 24-hour incubation period, the kinetic data should be extended to 24 hours, rather than truncated at 3 hours.
Author Response
We sincerely appreciate your constructive feedback and the time you have dedicated to reviewing this manuscript. We believe that our current dataset sufficiently supports our conclusions, as justified in our point-to-point reply. If you consider additional experimental data essential, we would be pleased to incorporate it. However, we would require additional time to conduct the necessary experiments.
Please find the detailed responses below and the corresponding corrections highlighted in red color in the re-submitted manuscript file.
Comment 1.1: The author found that 1 μg of S. copri OMVs contains 2 × 10⁸ OMV particles. To obtain 1 μg of S. copri OMVs, what culture volume of S. copri is needed?
Response 1.1: Thank you for this important comment. We realized S. copri liquid cultures in 250 mL volume to obtain approximately 100 mg of OMVs. We have now added this information in the manuscript (page number: 3, line: 116; page number 16, line 619-620).
Comment 1.2: Figures 1c and 1d require scale bars and labeling of the side panels for orthogonal projections of confocal sections.
Response 1.2: Thank you for your careful observation. As suggested, we have now included scale bars and labeling of side panels for orthogonal projections of confocal sections in Figures 1c and 1d.
Comment 1.3: Since Figures 1c and 1d show S. copri internalization, why only present OMV uptake kinetics and not S. copri uptake kinetics? For comparison, S. copri uptake kinetics should also be included.
Response 1.3: We appreciate your comment. However, the primary focus of this manuscript is on S. copri-derived OMVs. While we included DiO-labeled fixed S. copri as control for phagocytosis by macrophages, our study was not designed to compare uptake kinetics of whole S. copri bacteria with OMVs. Given the size of S. copri DSM 18205 of ~ 0.5 x 1 mm (Xiao et al., NPJ Biofilms Microbiomes. 2024;10(1):114), internalization is expected to occur predominantly via phagocytosis. In contrast, S. copri OMVs, which range from 47 to 253 nm, can also be internalized through other mechanisms. Correspondingly, we found that S. copri OMVs are primarily internalized by macropinocytosis and, to a lesser extent, by clathrin-dependent endocytosis.
Comment 1.4: Figure 1F shows that cytochalasin D and sucrose inhibit S. copri OMV uptake at different levels. It is worth investigating whether the combined use of these inhibitors can achieve complete blockage of S. copri OMV internalization. Furthermore, the effect of cytochalasin D and sucrose on S. copri cell internalization should be examined.
Response 1.4: Thank you for your valuable suggestion. The experiment presented in Figure 1f was designed to provide insight into endocytosis mechanisms involved in the uptake of S. copri OMVs by macrophages. Since we tested cytochalasin D and sucrose separately but not in combination (to achieve complete blockage of OMV internalization), we cannot exclude the possibility that other pathways, such as receptor-mediated endocytosis, caveolin-dependent or lipid raft-mediated uptake may also contribute to S. copri OMV internalization. We have now emphasized this point in the discussion (page 13 lines 442-446) of the manuscript.
Regarding the effect of cytochalasin D and sucrose on S. copri bacteria internalization, we would like to emphasize that our study focuses on S. copri OMVs. As explained in Response 1.3, we included DiO-labeled fixed S. copri as a control for phagocytosis by macrophages but did not aim to compare the uptake mechanisms between S. copri whole bacteria and OMVs, as these are expected to differ due to distinct size.
Comment 1.5: Figure 2. Samples were collected after 24 hours of incubation. Were earlier or later time points examined? Given that uptake kinetics were only measured up to 3 hours, does OMV uptake continue to increase over 24 hours? If uptake plateaus at an earlier time, would the same effects be observed as at 24 hours? Similarly, if uptake continues to increase, would a stronger response be observed?
Response 1.5: We selected the 24-hour time point, a standard in vitro incubation period, as the optimal duration to assess changes in M1/M2 marker expression and cytokine secretion in response to stimuli such as S. copri OMVs (Shiratori et al., Molecular Immunology 2017;88:58). Even though gene transcription and translation occur within minutes to hours, a longer incubation time is necessary for protein accumulation to detectable levels, turnover of existing proteins, cellular adaptation, and the establishment of complex regulatory networks. This also ensures a more uniform response across the entire cell population.
In this study, we did not examine earlier or later time points for the determination of M1/M2 markers and cytokines. However, based on our experience, a 24-hour incubation is required to observe a significant increase in M1 and M2 marker expression at protein level, whereas 4 to 6 hours of stimulation is insufficient to induce detectable changes.
Although we measured OMV uptake kinetics only up to 3 hours, macrophages continuously internalize extracellular material, including DiO-labeled OMVs, suggesting that uptake likely increases over 24 hours. However, for visualization by confocal microscopy, a 3-hour time point was optimal to detect DiO-labeled OMVs within a sufficient number of macrophages. Extending the observation period poses risks such as potential DiO redistribution to macrophage lipid structures (e.g., membranes), leading to false-positive signals, or OMV degradation within lysosomes.
Comment 1.6: While Pam3CSK4 effectively induces M1 polarization, the inclusion of a well-established M2 polarization agent as a positive control for M2 polarization would enhance the study's rigor.
Response 1.6: Thank you for this valuable comment. We used Pam3CSK4 primarily to confirm macrophage responsiveness to a proinflammatory stimulus via TLR1/2, rather than as a dedicated M1 polarization control. To investigate the effects of S. copri OMVs on different macrophage phenotypes, we induced M1 polarization using IFN-g and M2 polarization by IL-4. However, in contrast to Pam3CSK4, neither of the polarizing cytokines (IFN-g for M1 and IL-4 for M2) was sufficient to induce detectable cytokine secretion in macrophages, as demonstrated in the condition without stimulus in Figures 3, 4, and 5. In response to your suggestion, we have included a supplementary figure (Figure S4) demonstrating the upregulation of M1- and M2-associated markers in M1- and M2-polarized macrophages, respectively, compared to naïve M0 macrophages, which supports the validity of our polarization approach.
Comment 1.7: It appears S. copri OMVs have different effects on THP-1 cells and PBMC-derived macrophages. To resolve this discrepancy, a comparative analysis of uptake kinetics in both cell types is warranted. Figures 1e and 1f present data exclusively for THP-1 cells, corresponding results for PBMC-derived macrophages should be included. Considering the 24-hour incubation period, the kinetic data should be extended to 24 hours, rather than truncated at 3 hours.
Response 1.7: We appreciate the reviewer’s suggestion to conduct a comparative analysis of uptake kinetics between THP-1 cells and PBMC-derived macrophages. However, we would like to emphasize that our study employed THP-1-derived macrophages as a well-established and widely used model of human macrophages, as a first approach to study endocytosis, i.e. to establish optimal time points for visualization of OMV internalization by confocal microscopy and to evaluate kinetics and mechanisms of OMV uptake, particularly considering the limited number of primary macrophages that can be obtained from human heparinized blood samples.
Furthermore, our decision to measure uptake kinetics of DiO-labeled OMVs up to 3 hours is based on prior literature demonstrating that OMV internalization in macrophages occurs predominantly within the initial minutes to hours of exposure (O'Donoghue & Krachler, Cell Microbiol. 2016;18(11):1508–1517). Extending the analysis to 24 hours would not provide additional mechanistic insights, as intracellular accumulation of OMVs at later time points is likely confounded by factors such as cellular processing, degradation, and secondary effects rather than primary uptake kinetics.
Considering that the biological effect on marker expression and cytokine secretion is likely to show a delay with respect to receptor engagement by OMVs and/or OMVs uptake (as discussed in response 1.5), we suggest that the meaningfulness of extended internalization kinetic analysis beyond the early uptake phase until 24 h might be limited in the context of our research question.

Reviewer 2 Report
Comments and Suggestions for Authors
It is a good paper about Segatella copri outer membrane vesicles and its function. some suggests to be listed:
1 In title, logic shall be maked, I feel that it shall be changed to "outer membrane vesicles from Segatella copri are internalized by human macrophages to promote a pro-inflammatory response"
2,Line18-19: to be change "Segatella copri (S. copri) within the gut microbiota is associated with systemic inflammatory diseases,"
3 the logic and language shall be revised by senior expert.
4, it shall Emphasis on in vitro studies
5 the content in omv shall be analyzed
6 line 570, what is DSMZ?
7 line 612, THP-1 cell is , exposed to 100 nM phorbol 12-myristate 13-acetate (PMA), but PMA can harm the cells, and low concentration was used according to Standard method.
8. Infig4(F) and fig5(F), why IL10 is high after the treatment of OMV, when the macrophage is in M1 state? IL10 is a Anti-inflammatory factor
9, Pam3CSK4 as a TLR1/2 agonist induce different response to different polarizated macrophage, why?
10. Line 246, macrophages derived from THP-1 cell line have certain limitations, why did not use other macrophage Raw264.7 as a control?
11, CD86 as M1 marker shall be analyzed compared to CD206 as M2 marker
12 the count of M1 and M2 macrophage treated by OMV was nanlyzed by Flow cytometry
13, the consistency between THP-1 and MD cells shall be showed
Author Response
We sincerely appreciate your valuable feedback and the time you have taken to review our manuscript. We believe that the current dataset adequately supports our conclusions, as outlined in our point-to-point reply. However, if you consider additional experimental data essential, we would be pleased to incorporate it. As conducting these experiments would require additional time, we kindly note that an extension would be necessary.
Please find our detailed responses below, with the corresponding revisions highlighted in red in the resubmitted manuscript.
Comment 2.1: In title, logic shall be maked, I feel that it shall be changed to "outer membrane vesicles from Segatella copri are internalized by human macrophages to promote a pro-inflammatory response"
Response 2.1: Thank you for your suggestion. However, as we have not directly demonstrated in this manuscript that OMV internalization is a prerequisite for the pro-inflammatory effects of S. copri OMVs on human macrophages, we believe it is important to remain cautious in our conclusions. Therefore, we would prefer to maintain the original title to more accurately reflect our findings.
Comment 2.2: Line18-19: to be change "Segatella copri (S. copri) within the gut microbiota is associated with systemic inflammatory diseases,"
Response 2.2: Thank you for your suggestion. We changed phrasing to "Increased abundance of Segatella copri (S. copri) within the gut microbiota is associated with systemic inflammatory diseases, including rheumatoid arthritis." (page 1, line 18). This avoids overstating causality between S. copri and disease association, taking into account that increased S. copri abundance - rather than mere presence – may promote disease development.
Comment 2.3: the logic and language shall be revised by senior expert.
Response 2.3: As suggested, logic and language has now been revised by a senior expert.
Comment 2.4: it shall Emphasis on in vitro studies
Response 2.4: Thank you for your suggestion. We have emphasized the in vitro nature of our study more clearly in the revised manuscript (page 1, line 24; page 3, line 104; page 3, line 109; page 18, line 738)
Comment 2.5: the content in omv shall be analyzed
Response 2.5: Thank you for your valuable suggestion. We fully agree that characterizing OMV content is essential for understanding its proinflammatory effects on macrophages and elucidating the underlying mechanisms. To our knowledge, this is the first study investigating S. copri-derived OMVs. As an exploratory study, it focuses on describing the effects of S. copri OMVs on macrophages without claiming to reveal specific mechanisms of action or involved pathways. We have conducted preliminary mass spectrometry analysis of OMV protein content in our laboratory, identifying 71 proteins, though the function of many remains unknown and requires further investigation. We are currently working on a more detailed characterization of OMV protein and LPS content, which we plan to publish in a future study.
Comment 2.6: line 570, what is DSMZ?
Response 2.6: We appreciate the reviewer’s query. DMSZ refers to the Deutsche Sammlung von Mikroorganismen und Zellkulturen (German Collection of Microorganisms and Cell Cultures), a recognized biological resource center. ensure clarity and avoid any ambiguity, we have now included the full name in the manuscript (page 16, lines 594-596).
Comment 2.7: line 612, THP-1 cell is , exposed to 100 nM phorbol 12-myristate 13-acetate (PMA), but PMA can harm the cells, and low concentration was used according to Standard method.
Response 2.7: We used a moderate concentration of 100 nM PMA to induce THP-1 cell differentiation into macrophages, which falls within the commonly used range for in vitro studies on THP-1 macrophages (Genin et al., BMC Cancer 2015;15:577; Phuangbubpha et al., Cells 2023;12(19):1427). To mitigate potential cytotoxic effects, we restricted PMA exposure to 24 hours, followed by a 24-hour resting period in fresh RPMI medium. This approach ensures optimal macrophage differentiation while preserving cell viability.
As demonstrated in Supplementary Figure S3, THP-1 macrophages differentiated with 100 nM PMA exhibited viability above 90%, confirming minimal cytotoxicity. We have now incorporated this information into the manuscript (Page 17, Line 641-646).
Comment 2.8: Infig4(F) and fig5(F), why IL10 is high after the treatment of OMV, when the macrophage is in M1 state? IL10 is a Anti-inflammatory factor
Response 2.8: Thank you for your constructive comment. Many pro-inflammatory stimuli, such as LPS of Gram-negative bacteria – likely present in S. copri OMVs - induce not only a strong inflammatory response but also IL-10 production to counterbalance excessive inflammation and prevent tissue damage. Thus, even M1 macrophages are capable of producing IL-10 as part of a built-in control system to regulate immune responses. This phenomenon has also been described in M1 macrophages stimulated with OMVs from P. gingivalis (Cecil et al., Front Immunol. 2017;8:1017). We have included a more explicit explanation in the discussion (page 14, line 506-517).
Comment 2.9: Pam3CSK4 as a TLR1/2 agonist induce different response to different polarizated macrophage, why?
Response 2.9: Distinct polarization states of macrophages (M0, M1, M2) influence their expression of receptors, including TLR1/2, leading to differential activation of signaling pathways and cytokine production profiles in response to the TLR1/2 ligand Pam3CSK4 (Schlaepfer et al., J Virol 2014;88(17):9769–81; Quero et al., Arthritis Res Ther 2017;19:245; Wang et al., Front Immunol. 2014;5:614). Unpolarized naïve M0 macrophages express moderate levels of TLR1 and TLR2, resulting in an intermediate response to Pam3CSK4, as observed in their production of pro-inflammatory cytokines TNF-α and IL-6, alongside the anti-inflammatory cytokine IL-10 (Figure 3). In contrast, M1 macrophages, primed by IFN-γ, upregulate TLR2 and activate STAT1, NF-κB and MAPK signaling upon stimulation with Pam3CSK4, resulting in a strong pro-inflammatory response, characterized by high TNF-α and IL-6 production, with little to no IL-10, as demonstrated in Figure 4. M2 macrophages, induced by IL-4, generally exhibit lower TLR2 expression and favor anti-inflammatory signaling pathways involving STAT6. Consequently, they produce IL-10 while secreting reduced amounts of TNF-α and IL-6, as shown in Figure 5.
Comment 2.10: Line 246, macrophages derived from THP-1 cell line have certain limitations, why did not use other macrophage Raw264.7 as a control?
Response 2.10: This study specifically investigates the effect of S. copri OMVs on human macrophages. The RAW264.7 cell line is of murine origin, which limits its relevance for studying human immune responses. Furthermore, RAW264.7 macrophages do not exhibit clear M1/M2 polarization, making them less suitable for investigating macrophage plasticity compared to THP-1 cells. To address the limitations of the THP-1 cell line, we also used macrophages differentiated from human peripheral blood-derived monocytes for detailed in vitro studies on the effect of S. copri OMVs on M0, M1, and M2-polarized macrophages.
Comment 2.11: CD86 as M1 marker shall be analyzed compared to CD206 as M2 marker
Response 2.11: Thank you for your suggestion. However, we found CD80 more suitable to determine M1 differentiation. Thus, to address your comment, we included data on CD86 expression of macrophages as well as the ratio of CD80 to CD206 expression into the main Figures 2, 3, 4, and 5.
Comment 2.12: the count of M1 and M2 macrophage treated by OMV was nanlyzed by Flow cytometry
Response 2.12: We analyzed M1 and M2 macrophages 24 hours after treatment with OMVs by flow cytometry, acquiring approximately 10,000 cells per condition. Viability of human monocyte-derived macrophages was generally above 70% in M0, M1 and M2 macrophage populations. However, in M1 macrophages, treatment with S. copri OMVs at 1 mg/mL and 10 mg/mL reduced viability by approximately 20% (Supplementary Figure 3).
Comment 2.13: the consistency between THP-1 and MD cells shall be showed
Response 2.13: We appreciate the reviewer’s comment. The primary focus of this manuscript is to investigate the effect of S. copri OMVs on human macrophages rather than to directly compare responses of THP-1-derived macrophages and monocyte-derived macrophages (MDMs). In our study, we employed THP-1-derived macrophages as a widely used model for human macrophages to assess OMV internalization, uptake kinetics, endocytosis mechanisms, and their effects on macrophage phenotype and cytokine secretion. We acknowledge that THP-1 macrophages, as a tumor-derived cell line, have certain limitations, particularly in the expression levels of pattern recognition receptors and surface markers associated with M1/M2 polarization Shiratori et al., Molecular Immunology 2017;88:58). However, our findings demonstrate that the effects of S. copri OMVs were consistent between THP-1-derived macrophages and MD macrophages. Specifically, OMVs induced a shift towards an M1-like phenotype in both models, as evidenced by increased expression of M1-associated markers CD40 and CD80, decreased expression of M2 markers (CD206 in THP-1 macrophages and CD163 in MD macrophages), and stimulation of both pro-inflammatory (TNF-α) and anti-inflammatory (IL-10) cytokine secretion (Figures 2 and 3). It is important to note that while the overall trends were similar, the absolute expression levels of surface markers and TNF-α (but not IL-10) concentrations were lower in THP-1-derived macrophages compared to MD macrophages. This is likely due to the fact that THP-1 macrophages originate from a human leukemia cell line, which is inherently less immunogenic and exhibits a more limited immune response (Chanput et al., Int Immunopharmacol. 2014;23(1):37-45; Tedesco et al., Front. Pharmacol 2018;9).

Round 2
Reviewer 1 Report
Comments and Suggestions for Authors
All concerns have been addressed.